# Photoelectrochemical Performance of Strontium Titanium Oxynitride Photo-Activated with Cobalt Phosphate Nanoparticles for Oxidation of Alkaline Water

**DOI:** 10.3390/nano13050920

**Published:** 2023-03-01

**Authors:** Mabrook S. Amer, Prabhakarn Arunachalam, Mohamed A. Ghanem, Abdullah M. Al-Mayouf, Mark T. Weller

**Affiliations:** 1Chemistry Department, College of Science, King Saud University, Riyadh 11451, Saudi Arabia; 2K.A.CAR Energy Research and Innovation Center at Riyadh, King Saud University, Riyadh 11451, Saudi Arabia; 3Chemistry Department, Cardiff University, Cardiff CF10 3AT, UK

**Keywords:** oxynitride, photoelectrocatalysis, cobalt phosphate, oxygen evolution catalyst, water splitting

## Abstract

Photoelectrochemical (PEC) solar water splitting is favourable for transforming solar energy into sustainable hydrogen fuel using semiconductor electrodes. Perovskite-type oxynitrides are attractive photocatalysts for this application due to their visible light absorption features and stability. Herein, strontium titanium oxynitride (STON) containing anion vacancies of SrTi(O,N)_3−δ_ was prepared via solid phase synthesis and assembled as a photoelectrode by electrophoretic deposition, and their morphological and optical properties and PEC performance for alkaline water oxidation are investigated. Further, cobalt-phosphate (CoPi)-based co-catalyst was photo-deposited over the surface of the STON electrode to boost the PEC efficiency. A photocurrent density of ~138 μA/cm at 1.25 V versus RHE was achieved for CoPi/STON electrodes in presence of a sulfite hole scavenger which is approximately a four-fold enhancement compared to the pristine electrode. The observed PEC enrichment is mainly due to the improved kinetics of oxygen evolution because of the CoPi co-catalyst and the reduced surface recombination of the photogenerated carriers. Moreover, the CoPi modification over perovskite-type oxynitrides provides a new dimension for developing efficient and highly stable photoanodes in solar-assisted water-splitting reactions.

## 1. Introduction

The increased demand for energy worldwide requires an innovative approach to designing viable, economical, and effective systems for converting and storing renewable energy sources [1,2,3]. Hydrogen is the most abundant element and is regarded as the most promising clean renewable energy source. However, preserving a sustainable and renewable hydrogen energy source is challenging and requires breakthroughs in the production and storage of hydrogen and improvements in utilization. The solar photoelectrochemical (PEC) water-splitting reaction is considered the most favourable approach that can deliver a clean, economical, and domestically produced energy supply using abundant solar radiation [4]. Since Fujishima and Honda reported the TiO_2_ single crystal electrodes for solar water splitting schemes to produce hydrogen [1], numerous oxide-based materials have been studied [2,5,6]. In particular, photocatalytic semiconductors have been investigated for transforming sunlight into chemical energy through water splitting into solar fuels. To undertake effective water splitting, the photocatalysts must be chemically stable under the conditions of use and must possess a bandgap in the visible light region and energy level positions that are well-matched with the redox potentials of the water-splitting [7]. However, none of the examined materials (metal chalcogenides, metal oxides, and carbonaceous semiconductors) [8,9] are effective photocatalytic semiconductors for overall visible-light-driven water splitting. Notably, oxide semiconductors such as BiVO_4_ [10,11], Fe_2_O_3_ [12,13] and WO_3_ [14] have been extensively employed as electrodes to absorb the light photons and to permit adequate and stable PEC water-splitting reactions, but their conduction bands (CBs) are lower in energy than the redox potentials of the water [15]. Thus, research is ongoing to develop a semiconductor with appropriate CBs and valence band (VB) and a lower bandgap (<3 eV) for visible-light driving water-splitting reactions. The energy level edges of many metal oxides are not appropriate for absorbing the full range of visible-light energies, inhibiting the electrodes from being active across the whole visible-light region [16]. To promote the PEC features, it is vital to improve the lower energy in the semiconductor’s bandgap and enhance the light absorption across the visible part of the electromagnetic spectrum.

Oxynitride-based photoanodes are regarded as an alternative to oxide-based electrodes to grasp visible-light photons to produce solar fuels from water at the stoichiometric ratio [17,18]. In these photoelectrodes, atomic N substitution into the oxygen lattice sites shifts the VB potentials upwards via the hybridization of N 2p and O 2p orbitals thereby tuning the metal-anion covalency, band structure, and energy of the electronic states [19]. Furthermore, the CB minimum can be tuned with the N replacement because of the variation in anion electronegativity and/or lattice distortion [20,21]. However, once the N has been substituted into the equivalent oxide, the bandgap is shifted to the visible light region. Further, many electrode materials have been studied recently for water oxidation, without or with an externally applied bias [22,23] and different perovskite oxynitride semiconductors have been explored for PEC water oxidation schemes, such as LaTiO_2_N [24], ZnTaO_2_N [25], BaTaO_2_N [26], ZnLaTaO_2_N [27], SrTaO_2_N [28], LaTaON_2_ [29], and CaNbO_2_N [30]. Few photoelectrodes have achieved greater IPCEs (>20%) with a minimum applied potential and are comparatively weaker than the required thermodynamic potential for water electrolysis reactions [31]. However, the recombination of the photoinduced carriers and sluggish photon absorption efficiencies are the main obstacles of these electrodes, and these characteristics must be boosted for their use in water electrolysis reactions. Perovskite-based materials modified with suitable co-catalysts showed an innovative path toward the solar-assisted water-splitting reaction [32,33]. The introduction of water oxidation catalysts (WOCs) over oxynitride materials has shown a reduction in the recombination rate of excitons [34]. For instance, cobalt phosphate (CoPi) WOCs are exceptional materials for boosting the water electrolysis reaction at a moderately lesser potential [35,36]. Additionally, these electrodes tend to have comparatively higher abundance, self-healing individualities, and good efficacy under benign situations. CoPi co-catalyst sustains proton-coupled electron transfers during the water electrolysis, and the cyclic valency differs between Co(II/III) and Co(III/IV) redox couples [37] where the recombination rate of excitons reduces by increasing hole transport over the Co-ion valency cycle. Thus, this inspired the investigation of new oxynitride materials with CoPi as an effective approach for boosting solar-assisted water oxidation reactions.

Herein, we demonstrate the co-catalytic effect and PEC performance of CoPi-decorated SrTi(O,N)_3−δ_ (CoPi/STON) electrodes assembled by electrophoretic deposition and in situ photo-deposition process. It is exposed that the PEC performance of CoPi/STON films is enhanced considerably with the photo-deposition Co-Pi’s co-catalyst. Notably, the PEC enrichments are more predominant at the lower bias region, and approximately a four-fold improvement in photocurrent response is observed over the pristine film. The boosted PEC features of CoPi/STON can be attributed to the improvement in the charge separation and collection of holes produced in the electrode surface. In addition, these studies confirm a viable approach to improve energy conversion efficiency by merging water oxidation catalysts with innovative perovskite-type oxynitride electrodes.

## 2. Materials and Methods

### 2.1. Materials and Chemicals

The chemicals used in this study included sodium hydroxide (NaOH), strontium carbonate (SrCO_3_, 99%, Aldrich, Dorset, UK), titanium dioxide (TiO_2_, 99.8%, Alfa Aesar, Lancashire, UK), sodium chloride (NaCl, 99.7 Aldrich, Dorset, UK), ammonia (Electronic Grade, Air Products, Electronic Grade, Walton-on-Thames, UK), and (FTO) substrates.

The deionized water used in solution preparation was obtained from a Milli-Q ultrapure water purification system (resistance 18 Mohm).

### 2.2. Preparation of SrTi(O,N)_3−δ_ Powder

The strontium titanium oxynitride (STON, SrTi(O,N)_3−δ_) catalysts were prepared as described previously [38]. Firstly, the strontium titanate (SrTiO_3_) was initially prepared by a solid phase reaction of strontium carbonate (SrCO_3_^−^) and titanium dioxide (TiO_2_,) in the correct stoichiometric proportions at 1100 °C for 72 h with several regrinds. Then, the obtained white SrTiO_3_ was ground together with 50% wt.% of NaCl and treated with dry flowing gaseous of ammonia for 20 h at a flow rate of ca. 1 dm^3^/h and a reaction temperature of 850 °C. The mineralizers were leached from the fired strontium titanium oxynitride SrTi(O,N)_3−δ_ materials by washing thoroughly with an excess of deionized water to obtain the pure strontium titanium oxynitride catalyst.

### 2.3. Preparation of CoPi-Modified STON Photoanodes

The photoanode of the STON catalyst was obtained via an electrophoretic deposition (EPD) approach. Briefly, 20 mg of the SrTi(O,N)_3−δ_ catalyst was mixed with 10 mg of iodine in 25 mL of acetone and then subjected to ultrasonication for 1.0 h to acquire homogenously dispersed oxynitride particles. The EPD was performed between two parallel fluorine-doped tin oxide (FTO) substrates (1 × 2 cm^2^) kept in the homogenously dispersed oxynitride solution at a distance of 1.0 cm. The oxynitride materials were electrochemically deposited over the negative FTO electrode by applying a bias of 10 V for 3 min between the FTO substrates. Finally, the photoanodes were calcined at 350 °C for 1.0 h under N_2_ flow. The modification of the SrTi(O,N)_3−δ_ photoanode with cobalt phosphate (CoPi) was performed by the in situ photo-deposition in a reactor containing 10 mL of 0.1 M potassium phosphate solutions (pH 7.0) and 0.5 mM CoCl_2_. Subsequently, the acquired solution was irradiated by UV light (400 W Xe lamp) and the CoPi decorated STON/FTO photoanode with varied CoPi loading was obtained by varying the duration of the photo-deposition process. The pristine SrTi(O,N)_3−δ_ and CoPi modified SrTi(O,N)_3−δ_ photoanodes were labelled hereafter as STON and CoPi/STON, respectively.

### 2.4. Materials Characterization

The crystallinity and phase of the fabricated STON and CoPi/STON photoanode were investigated using powder X-ray diffraction (XRD, Rigaku Miniflex-600) with Cu-K_α_ radiation (λ = 1.5418 Å). The surface morphology and EDX elemental analysis were performed using a scanning electron microscope (SEM, S-4800, Hitachi, Thermo Fisher Scientific, Oxford, UK) fitted with an EDX spectrometer (EX350, Horiba). The optical properties of the fabricated photoanodes were measured using a UV-vis-NIR spectrophotometer (Shimadzu UV-2600, Kyoto, Japan) and the thickness of the obtained electrode films was determined through Bruker Dektak XT profilometer.

### 2.5. Photoelectrochemical (PEC) Performance Measurements

Water oxidation using pristine STON and CoPi/STON photoanodes as working photoelectrodes were evaluated in a traditional three-electrode quartz cell in conjunction with a potentiostat (BioLogic SAS, VSP-0478, Paris, France) under on/off AM 1.5 G simulated sunlight (Xenon lamp (300 W, HAL-320)). The platinum sheet (Pt, geometric area 1.0 × 1.0 cm^2^) and an Ag/AgCl electrode were employed as counter and reference electrodes, respectively. The PEC measurements were conducted in a 0.5 M sodium hydroxide (NaOH) solution (pH~12.7) and the linear sweep voltammograms (LSVs) were acquired within the applied bias range from −0.6 to 0.6 V vs. Ag/AgCl at the sweep rate of 5 mV/s at 25 °C in an air-saturated solution. All PEC measurements were attained after stabilizing the photoelectrode by performing seven cyclic voltammetry runs under dark conditions. A Mott-Schottky (M-S) plot was obtained at a frequency of 1.0 kHz in the dark using an electrochemical impedance analyzer (EIS, BioLogic SAS, VSP-0478) and EIS analysis was carried out in the dark and under illumination at an applied bias potential of 0.3V vs. AgAg/Cl with an AC voltage magnitude of 20 mV.

## 3. Results and Discussion

### 3.1. Physicochemical Characterization of the SrTi(O,N)_3−δ_ and CoPi/SrTi(O,N)_3−δ_ Catalysts

The strontium titanium oxynitride (SrTi(O,N)_3−δ_) surface morphology as examined by scanning electron microscope (SEM) is shown in Figure 1. The SEM images of the surface morphology at a different magnification of the pristine SrTi(O,N)_3−δ_ (Figure 1a,b) show a cubic shape crystal structure with various micro-sizes and sharp crystal edges and faces, demonstrating a higher crystalline nature of the SrTi(O,N)_3−δ_ catalyst.

On the other hand, the SEM images in Figure 1c,d of CoPi modified SrTi(O,N)_3−δ_ prepared at photo-deposition time of 30 min revealed the SrTi(O,N)_3−δ_ crystal uniformly decorated with CoPi nanoparticles with an average particle size of 15 nm, as shown in the inset histogram in Figure 1d. Interestingly, the CoPi nanoparticles are homogenously covered by the SrTi(O,N)_3−δ_ surface without obvious aggregation which is expected to facilitate the process of photoinduced hole generation at the active reaction sites [38].

Figure 2 shows the energy-dispersive X-ray spectroscopy (EDX) analysis assessing the elemental composition of the modified CoPi/STON photoanode. The elemental analysis mapping as shown in Figure 2 disclosed the homogenous distribution of the Sr, Ti, Co, O, N, and P elements throughout the CoPi/STON film.

In addition, the elemental mapping analysis of pristine SrTi(O,N)_3−δ_ catalyst is shown in Appendix A, which reveals a homogenous distribution of Sr and Ti, O, and N in the electrode films. The EDX elemental analysis shown in Figure 2h and Table 1 reveal the cobalt content equals 0.22 wt.% in the case of CoPi/STON photoanode, which indicates that a tiny amount of CoPi has been deposited on the surface of SrTi(O,N)_3−δ_ catalyst. In contrast, for pristine SrTi(O,N)_3−δ_ electrodes, Appendix A shows the EDX elemental analysis recorded 21.70, 27.85, 44.52, and 5.86 wt.% for Sr, Ti, O, and N elements, respectively, that are impeccably equivalent to the chemical composition of Sr_0.3_Ti_0.7_O_2.7_N_0.4_ considering the error in EDX analysis of ±0.01%. This EDX mapping examination primarily discloses a homogenous CoPi deposition over the SrTi(O,N)_3−δ_ surface, indicating the successful modification of oxynitride perovskite without apparent aggregation.

Figure 3 shows the XRD pattern for both the pristine SrTi(O,N)_3−δ_ powder and the modified CoPi/STON (deposition time 30 min) which follows the pure SrTiO_3_ perovskite structure with a cubic perovskite crystal structure crystallizing in the space group Pm-3m (JCPDS No. 01-084-0443). Moreover, the background peaks corresponding to the conductive FTO substrate (JCPDS No. 41-1445) can be observed in the case of the CoPi/STON photoanode (blue line).

As shown by the XRD pattern, the perovskite structure of SrTi(O,N)_3−δ_ powder was maintained upon doping of the nitride anion (N^3−^) and the estimated lattice parameters of the SrTi(O,N)_3−δ_ catalyst showed an expanded value of 3.905 Å compared to the SrTiO_3_ lattice parameter of 3.886 Å, confirming the incorporation of the nitride ion and variance in the oxygen vacancies due to the partial reduction in Ti^4+^ to the larger Ti^3+^ [39]. After the deposition of CoPi on the SrTi(O,N)_3−δ_ electrode at all deposition times, no noticeable changes can be observed in the pattern of the CoPi/STON compared with that of pristine SrTi(O,N)_3−δ_ powder as shown in Appendix A, indicating that the SrTi(O,N)_3−δ_ catalyst maintained their crystal structure after the electrophoretic deposition of CoPi nanoparticles.

The surface composition features of the pristine SrTi(O,N)_3−δ_ and modified CoPi/STON composite electrodes were investigated by XPS analysis, as shown in Figure 4. The survey XPS spectrum of pristine SrTi(O,N)_3−δ_ as well as the CoPi/STON composite (30 min. of CoPi deposition) electrodes revealed the existence of all elements of Sr, Ti, Co, P, O, and N without obvious contaminations (Figure 4a). The Sr 3d core spectra of the SrTi(O,N)_3−δ_ and CoPi/ STON composite electrodes (Figure 4b) displayed two main peaks of Sr 3d_5/2_ and Sr 3d_3/2_ components at 133.6 and 135.2 eV, respectively. They are attributed to the bonds of Sr atoms in the perovskite structure of SrTi(O,N)_3−δ_. Further, Figure 4c presents the deconvolution of the Ti 2p spectrum of both SrTi(O,N)_3−δ_ and CoPi/ STON composite electrodes. The Ti 2p spectra show the characteristic doublet of spin–orbit coupling (2p_3/2_, 2p_1/2_) for both photoanodes, with the highest intensity peak of the Ti 2p_3/2_ component at approximately 459.3 eV binding energy (BE) due to the presence of Ti^4+^ ions in and a lower intensity peak at ca 460.10 eV due to the existence of Ti^3+^ ions in SrTi(O,N)_3−δ_ [40,41]. Figure 4d shows the core O1s spectra of both photoanodes with the high-resolution O 1s spectra showing a more substantial narrow peak at ~530.44 eV and a broad peak at around 531.80 eV, implying the existence of two dissimilar O chemical states of birding oxygen in the Ti–O bond and the non-stoichiometric oxygen atom (vacancy) in the Ti–O–N or Ti–N–O bonds, respectively [42,43]. In addition, the corresponding N 1s deconvoluted spectrum of both photoanodes is shown in Figure 4e. The oxidation states of nitrogen in oxynitride compounds can be varied from N^3–^ to N^3+^ and the main deconvoluted peak of STON spectra at higher BE of 401.0 can be related to the N–Ti–O and/or Ti–O–N–O oxynitride bonding of positively charged species of N^1+^, N^2+^, and N^3+^. On the other hand, the lower intensity peak around BE 397.0 eV can assign to the negatively charged species of N^3−^, N^2−^, and N^1−^ species in various combinations of N–Ti–O oxynitride bonding [44]. In the case of CoPi/STON film, it is noted that the peak area of negatively charged N species (at (397.0 eV) is considerably decreased compared to the peak area of positively charged species (at 401.0 eV), suggesting that the CoPi nanoparticles are preferentially photo-deposited at the negative nitrogen sites of the oxynitride substrate. Figure 4f shows the deconvoluted XPS peaks of Co 2p of CoPi/STON photoanode which reveal two main bands of 2p_3/2_ and 2p_1/2_ components around 784.6 and 799.8 eV, respectively, that correspond to the Co^2+^ ion bonding in deposited CoPi co-catalyst [45]. The XPS core spectra of P 2p of CoPi/STON photoanode are shown in Figure 4g, where two well-resolved peaks are observed at BE of ~133.6 and 135.3 eV, representing the P 2p_3/2_ and P 2p_1/2_ components, respectively, of the P atom in the phosphate group, endorsing that P subsists as the nature of the phosphate group.

The UV–vis absorption spectroscopy of the samples was employed to characterize the electronic transitions and band gap energies of the SrTi(O,N)_3−δ_ films. Figure 5a shows the optical absorption plots of both pristine STON film and the CoPi/STON films, and the bandgap values were calculated according to the Tauc-plot function as shown in Figure 5b. The STON has a pale blue colour with strong absorption waves within the range of 300 to 800 nm wavelength zones. The blue colour of the STON may originate from the reduction in Ti^4+^ to Ti^3+^ cations due to the reductive atmosphere of ammonia, as ordinarily, SrTiO_3_ is white [38]. However, in the case of CoPi/STON, electrode films show different absorption profiles with the absorption edge point significantly shifted towards the lower wavelengths compared with that of pristine STON. The band gap was estimated via the Tauc plot as seen in Figure 5b, where the STON and CoPi/STON films show estimated bandgap energy of 1.7 and 1.79 eV, respectively.

### 3.2. Photoelectrochemical (PEC) Performance of the SrTi(O,N)_3−δ_ Photoanodes

Initially, the comparative electrochemical surface area (ECSA) of the STON and modified CoPi/STON photoanodes was assessed by the capacitive region of the cyclic voltammogram (CV)s. Appendix A shows the CV plots performed at different sweep rates in 0.5 M NaOH solution (pH~12.5). The ECSA as assessed by capacitive current and sweep rate dependence of CoPi/STON was about eight times higher than that of the pristine SrTi(O,N)_3−δ_ electrode, which indicates a significant enhancement in the active site and ECSA upon the deposition of CoPi over the surface of SrTi(O,N)_3−δ_ electrode.

The PEC features of the SrTi(O,N)_3−δ_ electrode films were examined and optimized using the linear sweep voltammogram (LSV) and chronoamperometric (CA) measurements. Figure 6a displays the LSV tests under chopped illumination of AM 1.5 G recorded at 10 mV/s in 0.5 M NaOH for STON film deposited on FTO substrate by electrophoretic deposition at different deposition times. The results show that the current density of the photoanode changes dramatically as the deposition time increases. As shown in Figure 6a, the maximum current density (5.6 μA/cm^2^ at 1.25 V vs. RHE) was achieved at around 3.0 min deposition time of loaded SrTi(O,N)_3−δ_ on FTO (equivalent to the loading of 1.5 mg and a thickness of 1.6 μm). At greater deposition time (>3 min), the photocurrent density considerably reduced, which could be attributed to a thick film being formed and the photoinduced holes being transported within several SrTi(O,N)_3−δ_ particles at the catalyst/electrolyte interface, and consequently, slower kinetics of the hole transfer and the reduced photocurrent is noticed [46]. The optimization of front- and back-side irradiation is achieved by LSV plots of STON photoanodes, as shown in Figure 6b. The pristine STON photoanode exhibits a more superior photocurrent via back irradiation than the front one, which is consistent with the earlier works [36] and can be credited to the photoinduced electrons travelling a longer distance before being collected by FTO back contact during the front-side illumination. Figure 6c displays the photocurrent transient under AM 1.5 G continuous dark and light illumination for pristine STON electrodes as measured in 0.5 M NaOH (pH 12.5). The maximum performance with a photocurrent density of 5.6 μA/cm^2^ at 1.25 V vs. RHE was recorded for the STON photoanodes with a thick layer of about 1.6 μm, and under back-illumination, found to be the optimum condition to ensure faster water oxidation kinetics over the electrode surface and adequate transport of photoinduced carriers of SrTi(O,N)_3−δ_ particles and the conducting substrate.

The in situ photochemical deposition of CoPi over SrTi(O,N)_3−δ_ particles under illumination conditions were established to boost the water oxidation reaction at the electrodes by reducing the electron-hole recombination [46,47]. Here, the attractive CoPi water oxidation catalyst has been applied onto SrTi(O,N)_3−δ_ particles photoanode by in situ photo-deposition method. To witness the influence of photo-assisted decoration of SrTi(O,N)_3−δ_ with CoPi on the generated photocurrent, the current-voltage analysis of CoPi/STON photoanodes was assessed under visible light conditions. More importantly, it must be validated that the level of the CoPi loading depends mainly on the fabricated electrode’s morphological features, time, and methodology of CoPi decoration. Under our experimental conditions, Figure 7a shows the optimal time intervals for CoPi photo-deposition over electrodes were assessed to be about 30 min, as explored by the improved photocurrent of the SrTi(O,N)_3−δ_ photoanode decorated with CoPi cocatalyst and obtained at different deposition times. Figure 7b shows the plot of the relationship between the CoPi/STON photocurrent measured at 1.25 V vs. RHE and the CoPi deposition time. It shows that the thicker CoPi incorporation into SrTi(O,N)_3−δ_ at deposition time higher than 30 min results in a decrease in photocurrent. This can be clarified as the electrocatalytic water oxidation reaction is happening at the CoPi/oxynitride interface and when the CoPi layer becomes denser, the photoinduced holes could be transferred in between many CoPi molecules and the CoPi/electrolyte interface, which results in slow hole transfer, and consequently, a decrease in the obtained photocurrent [48]. In contrast, the thin CoPi layer of linked cobalt ions on SrTi(O,N)_3−δ_ surface is easy to acquire the photo-holes to produce the active catalytic species of cobalt (Co^4+^) required for water oxidation [49]. In comparison, the PEC performance of CoPi/STON photoanode displayed about 400 mV cathodic shift of the photocurrent onset potential and an enhancement in the photocurrent of nearly 26.0 μA/cm^2^ at 1.25V vs. RHE which is more than four times higher compared to the pristine STON electrode (5.6 μA/cm^2^ at 1.25 V vs. RHE). The obtained results validate that the photocurrent enhancement observed after modified CoPi/STON films is significant, which might be attributed to catalytic features of CoPi active sites towards water oxidation consistent with that reported literature [49,50,51,52].

The PEC performance of STON and the modified CoPi/STON anodes were further improved by employing the Na_2_SO_3_ as an effective hole scavenger in an alkaline solution. Figure 8a shows typical LSV measurements at 20 mV/s for STON and CoPi/STON photoanode under dark and visible-light conditions in the presence of 0.5 M Na_2_SO_3_ as a hole scavenger in 0.5 M NaOH (pH~12.5). The addition of Na_2_SO_3_ as a hole scavenger to the alkaline solution results in a significant enhancement (more than five times) of the water oxidation photo-current at both CoPi/STON and pristine STON photoanodes with a recorded photocurrent of 138.7 and 31.2 µA/cm^2^ at 1.25 V vs. RHE, respectively. Moreover, in the presence of Na_2_SO_3_, the CoPi/STON exhibited a remarkable negative shift in the onset potential (more than 300 mV), indicating the improvement of the water photo-oxidation reaction at a lower potential.

Notably, the presence of Na_2_SO_3_ (electron donator) remarkably enhanced the PEC performance of both CoPi-modified and pristine SrTi(O,N)_3−δ_ electrodes towards the water photo-oxidation reaction via suppressing the recombination of charge carriers and improving the injection of photoinduced holes into the CoPi/STON/electrolyte junction.

Figure 8b shows the long-term durability of the STON and modified CoPi/STON photoanode was relatively assessed in the 0.5 M NaOH solution for more than 3 h at a constant potential of 1.25 V vs. RHE under on/off dark and light illumination in 0.5 M NaOH electrolyte. Particularly, the obtained photocurrent values are consistent with the results attained from the equivalent LSV curves shown earlier (Figure 6a and Figure 7a). It was found that in both cases of STON and modified CoPi/STON photoanodes, the photocurrent has been gradually increased by more than 40% to reach a value of about 17.0 and 36.3 µA/cm^2^, respectively, at the end of three hours of the photo-electrolysis process. This is presumably due to the convection effect of evolved oxygen gas as well as the catalyst surface activation. This signifies the obtained photoanodes have considerable long-lasting durability and photo-corrosion resistance during the electrolysis period. In addition, the photocurrent was quickly retained with an even slight enhancement under chopped light conditions, confirming the modification of SrTi(O,N)_3−δ_ with CoPi nanoparticles boosted both the PEC features and the durability of the photo-electrodes [12].

The surface morphology, crystal structure, and elemental analysis of the modified CoPi/STON photoanodes after the stability test (3 h electrolysis) were investigated and the results are provided in Appendix A and Table 1. Interestingly, as shown in Appendix A, the CoPi/STON photoanode had no obvious surface morphology changes after being used for three hours in the electrolysis test. However, the deposited CoPi nanoparticles become less defined and observable in comparison to the SEM images in Figure 1c,d of the fresh CoPi/STON photoanode. Moreover, as shown in Appendix A, the XRD of the used CoPi/STON photoanode matched very well with that of the fresh CoPi/STON electrode, confirming the crystal structure of the pristine STON substrate after being used in the electrolysis test was maintained. However, as shown in Table 1 above, the elements wt.% of the CoPi/STON electrode has been significantly changed after being used. The oxygen content significantly increased from 48.81 to 65.02 wt.% due to the electrochemical oxidation during the electrolysis in an alkaline solution. On the other hand, the elements of Sr, Ti, Co, and P have significantly decreased from 19.01, 27.06, 0.22, and 1.04 to 12.74, 21.77, 0.12, and 0.35, respectively. This indicates the leaching of the elements from the CoPi/STON surface during the stability electrolysis process, even though the CoPi/STON electrode performance (photocurrent) is increased by more than 40% at the end of the long-term stability test as shown above in Figure 8b.

The semiconductor’s band positions and charge carrier mobility are essential to understanding the PEC features of photoelectrode materials. Here, the Mott-Schotkky (M-S) method and Nyquist plot measurements are performed to understand the PEC features of the STON and modified CoPi/STON photoanodes. Figure 9a shows the Nyquist measurements of both STON and modified CoPi/STON photoanodes examined under the dark and light irradiation conditions at 1.25 V vs. RHE and the inset shows the equivalent circuit. Moreover, the same data is plotted in the Bode format, as seen in Figure 9b, and the overall obtained impedance parameters of resistances, constant phase element (CPE), Flat band potential, and donor densities values, obtained from the Nyquist plots under an equivalent circuit model shown in the inset of Figure 9a and Mott Schottky analysis, are summarized in Table 2. Within the lower frequency zone and corresponding equivalent circuit consistent with the examined catalysts’ charge transfer resistance (R_2_), the CoPi/STON electrodes were best fitted to a one-RC circuit model, which holds one resistor, one RC circuit, and one Warburg element. The radius of the arc of the Nyquist plots of the CoPi/STON electrode films is smaller than for the pristine STON electrode, demonstrating rapid interfacial charge transfer and also prominent in separating photoinduced carriers. In the case of the CoPi/STON electrode, the charge transfer resistance was reduced, and the capacitance upsurged, validating that the CoPi-incorporation improved the charge separation at the bulk photoanode and enhanced their PEC water photo-oxidation performance. Additionally, the CoPi decorated SrTi(O,N)_3−δ_ surface decreased the charge transfer resistance and increased the capacitance at the CoPi-loaded SrTi(O,N)_3−δ_ /electrolyte interface. Consequently, we can confirm that CoPi incorporation enhances its behaviour by assisting the charge separation and water oxidation reaction at the SrTi(O,N)_3−δ_ electrode surface, which is consistent with the results obtained for CoPi/BiVO_4_ and CoPi/TiO_2_ reported earlier in the literature [49,50,51,52,53].

Figure 9c shows the M-S experiments executed in 0.5 M NaOH solution (pH = 12.5) at 100 Hz in dark conditions. It can be seen in Figure 9c that the intercept on the potential axis of STON photoanodes shows a flat band potential (E_FB_) is acquired at −0.035 V vs. RHE at pH = 12.5 while the intercept of the potential axis of the CoPSTON electrode reveals an E_FB_ obtained at 0.100 V vs. RHE. More importantly, the CoPi/STON exhibits a more positive E_FB_ than the pristine STON electrode. Moreover, the donor density of the CoPi/STON electrode was estimated to be 2.82 × 10^21^ cm^−3^, which is much higher than that of pristine STON (estimated at 7.7 × 10^20^ cm^−3^), which signifies the effectiveness of CoPi modification for improving the electrical conducting property of the SrTi(O,N)_3−δ_ electrode by increasing its charge carrier density. These results validate that incorporating CoPi reduced charge carrier recombination and assisted the water electrolysis to happen at a lower bias region by varying the reaction pathway [49,54]. Moreover, the decorated CoPi films are thin enough to provide a rapid output of the photoinduced holes from SrTi(O,N)_3−δ_ catalyst to water oxidation, resulting in the relieved charge accumulation at the oxynitride/electrolyte interface.

Finally, a possible PEC splitting mechanism of CoPi/STON is proposed in Figure 10. In the case of the CoPi/STON interface, the PEC performance is boosted by the incorporated CoPi that provides the active sites for OH^−^ oxidation to O_2_ as well as suppresses the charge recombination by consuming the photogenerated holes while the electrons can reach the Pt electrode to reduce water under the externally applied bias. Furthermore, the Mott–Schottky results demonstrated a substantial difference after CoPi introduction over SrTi(O,N)_3−δ_ surface, thus promoting charge separation efficiencies extraordinarily. The CoPi cocatalysts, as a greatly effective catalytic material for oxygen evolution reaction, enhanced the hole-trapping features to accelerate the separation of photogenerated electron-hole pairs. Profiting from the self-circulation of CoPi, in this favourable situation, photoinduced charge carriers are well separated and have adequate time for water oxidation, which boosts the PEC performance of the CoPi/STON than pristine SrTi(O,N)_3−δ_ film. These results contribute to the understanding and designing of effective PEC cells for solar fuel production.

## 4. Conclusions

The SrTi(O,N)_3−δ_ photocatalyst with anion vacancies was obtained by solid-state reaction and reduction in an ammonia atmosphere. Photoanodes of pristine SrTi(O,N)_3−δ_ and CoPi photo-activated SrTi(O,N)_3−δ_ were assembled and the PEC performance was examined for alkaline water oxidation. The PEC performance of SrTi(O,N)_3−δ_ film is remarkably enhanced after photo-deposition of CoPi co-catalyst, particularly in the lower bias region and in presence of a Na_2_SO_3_ hole scavenger. In an alkaline sulphite solution, the CoPi/STON achieved a photocurrent density of 138.7 µA at 1.25 V vs. RHE, which is more than four times higher compared to the pristine SrTi(O,N)_3−δ_ electrode (31.2 µA/cm^2^) electrode. Moreover, the layer of CoPi nanoparticles on SrTi(O,N)_3−δ_ surface significantly increased the charge carrier density and reduced the charge recombination during the alkaline water oxidation process. The CoPi/ SrTi(O,N)_3−δ_ photocatalyst with a bandgap < 2.0 eV can function as a photoanode for water oxidation reaction under visible light illumination with superior performance than the pristine SrTi(O,N)_3−δ_.

## Figures and Tables

**Figure 1 nanomaterials-13-00920-f001:**
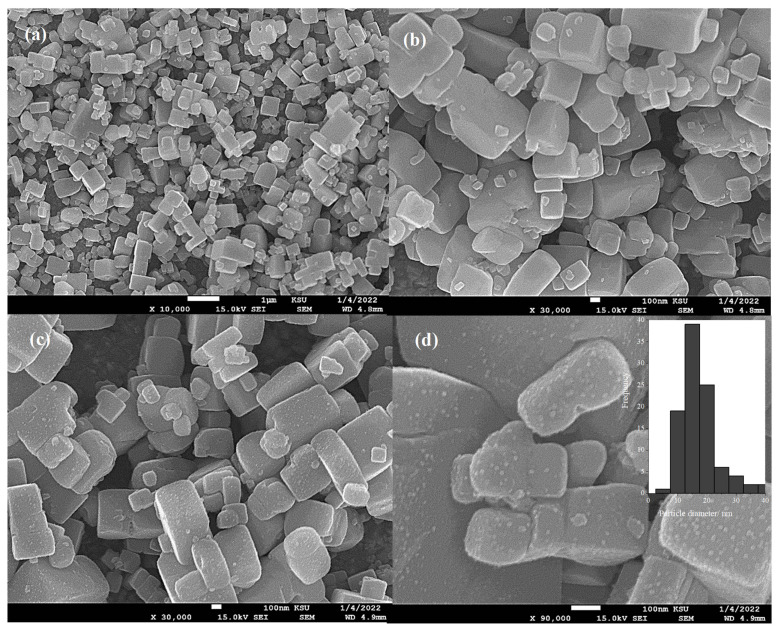
Scanning electron microscopy photographs at (**a**) low, (**b**) higher magnification of pristine SrTi(O,N)_3−δ_, (**c**) low and (**d**) high magnification of CoPi decorated SrTi(O,N)_3−δ_ photoanode obtained through electrophoretic and photo-deposition process onto FTO for 30 min.

**Figure 2 nanomaterials-13-00920-f002:**
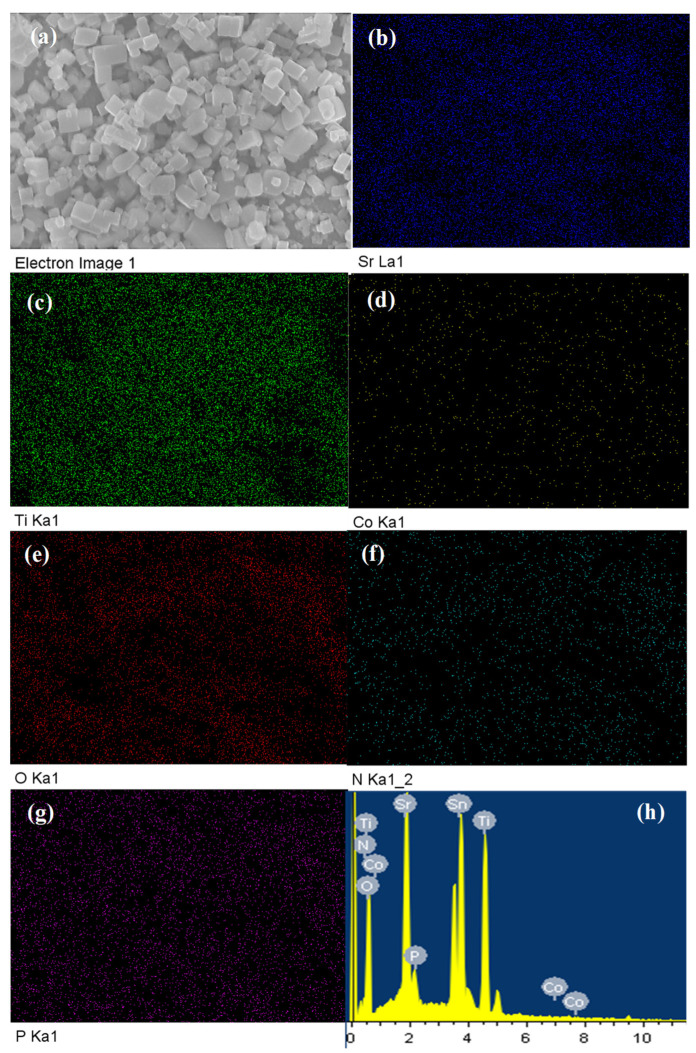
EDX elemental mapping analysis of CoPi/STON electrode, (**a**) SEM electronic image, (**b**–**g**) the distribution map of Sr, Ti, O, N, Co, and P elements, respectively, and (**h**) the energy-dispersive X-ray (EDX) profile of CoPi/STON film.

**Figure 3 nanomaterials-13-00920-f003:**
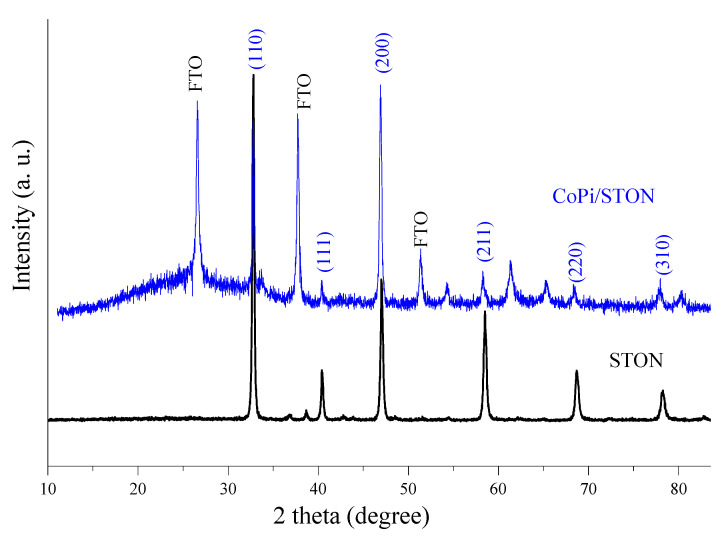
Normalized XRD patterns of pristine SrTi(O,N)_3−δ_ (STON) powder and CoPi/STON film deposited on FTO substrate.

**Figure 4 nanomaterials-13-00920-f004:**
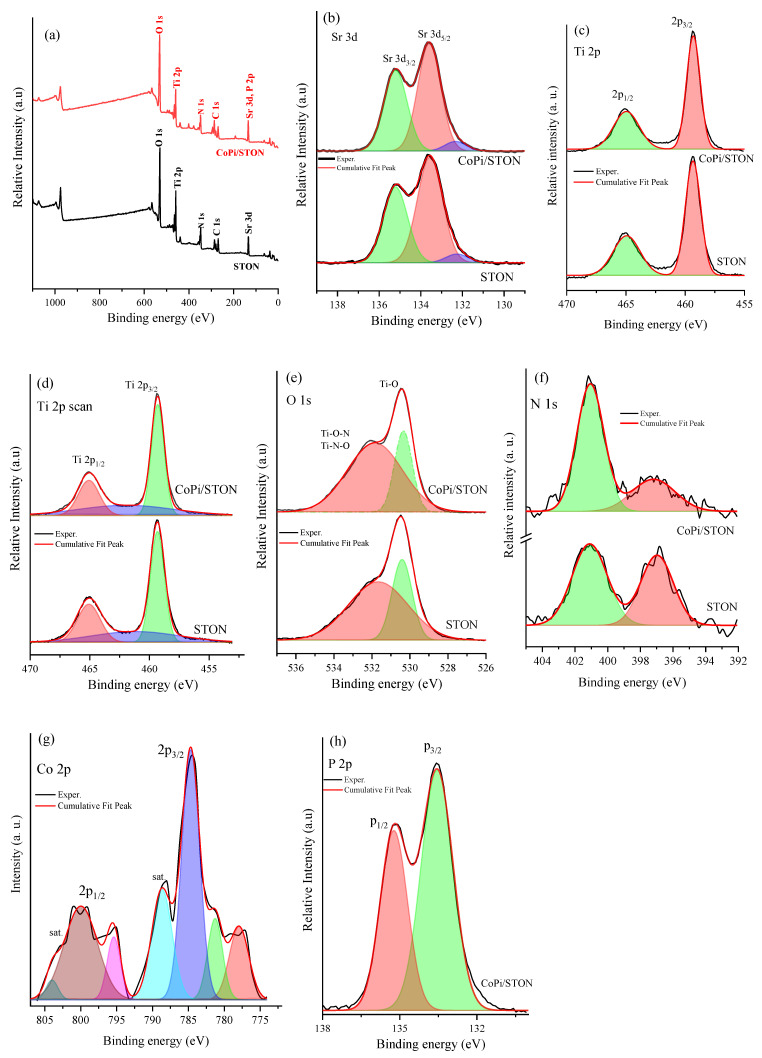
(**a**) Overall XPS-survey spectrum of pristine SrTi(O,N) and modified CoPi/SrTi(O,N) after 30 min deposition films, (**b**) Sr 3d, (**c**) Ti 2p, (**d**) Ti 2p, (**e**) O 1s, (**f**) N 1s, (**g**) Co 2p, and (**h**) P 2p core scan.

**Figure 5 nanomaterials-13-00920-f005:**
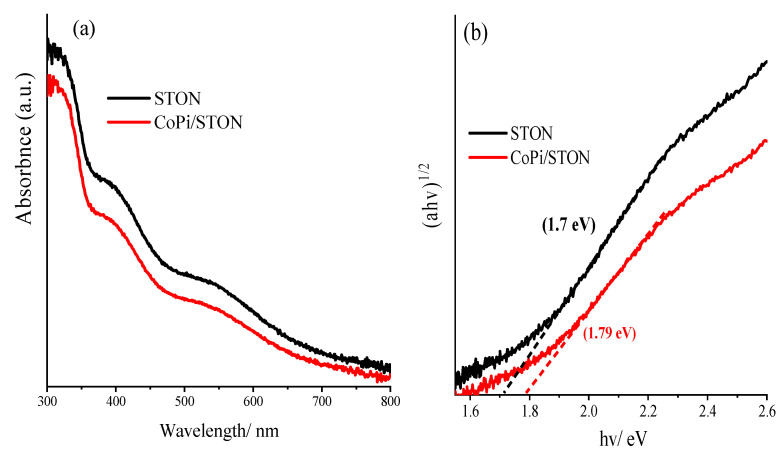
(**a**) UV-Vis absorbance spectra of perovskite-type titanium oxynitride of pristine SrTi(O,N)_3−δ_ (STON) and CoPi/ STON photoanodes, (**b**) the corresponding Tauc plot and the energy gap of SrTi(O,N)_3−δ_ and CoPi/STON photoanodes.

**Figure 6 nanomaterials-13-00920-f006:**
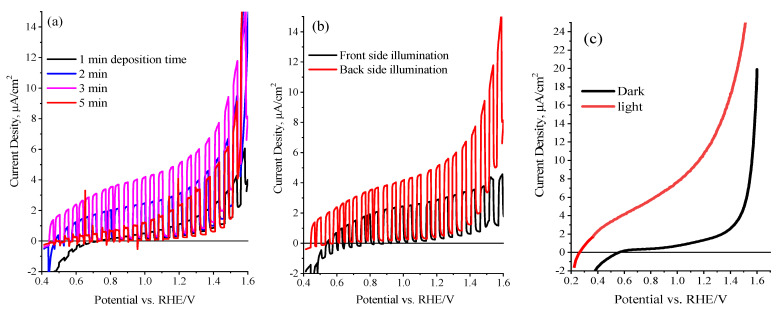
(**a**) Current-voltage plots in aqueous 0.5 M NaOH (pH 12.5) under intermittent visible-light illuminations for SrTi(O,N) films prepared at different electrophoretic deposition times. (**b**) Comparative plots on the variation in the front and back-side illumination. (**c**) LSV under dark and light continuous illumination for SrTi(O,N) photoanode.

**Figure 7 nanomaterials-13-00920-f007:**
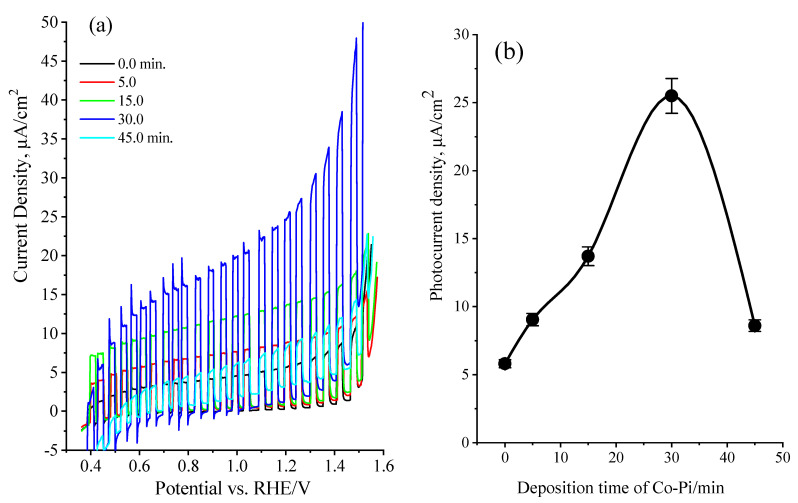
(**a**) LSV curves in aqueous 0.5 M NaOH solution (pH 12.5) under visible-light excitation for SrTi(O,N), and that modified with CoPi co-catalyst photo-deposited at different deposition times, (**b**) the plot of the relationship between the CoPi/SrTi(O,N) photocurrent measured at 1.25 V vs. RHE and the CoPi deposition time.

**Figure 8 nanomaterials-13-00920-f008:**
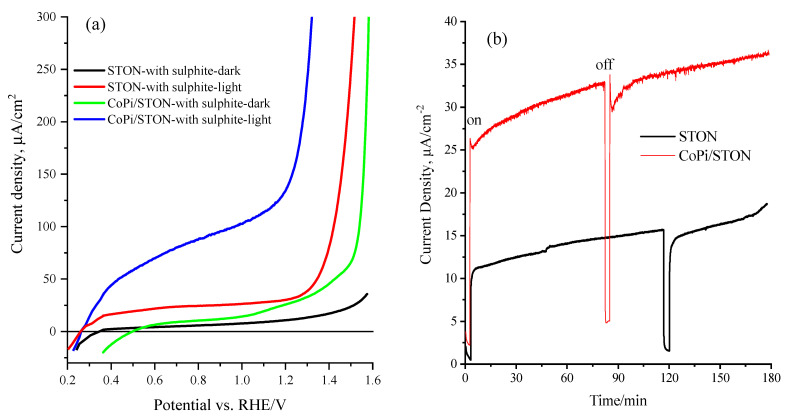
(**a**) Linear Sweep voltammograms at 20 mV/s of STON and modified CoPi/STON photoanodes in the presence of 0.5 M Na_2_SO_3_ and 0.5 M NaOH under continuous dark and light illumination, (**b**) long-term chronoamperometry of STON and modified CoPi/STON photoanodes at an applied potential of 1.25 V vs. RHE and under on/off dark and light illumination in 0.5 M NaOH electrolyte.

**Figure 9 nanomaterials-13-00920-f009:**
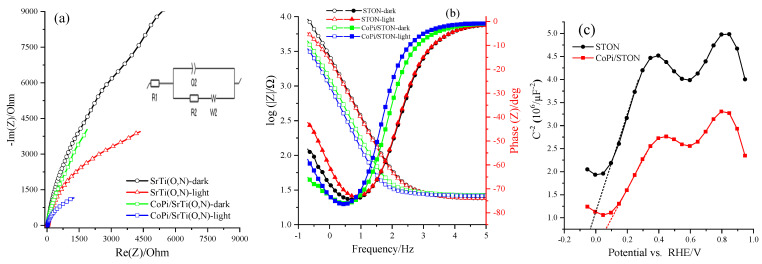
(**a**) Nyquist plots, (**b**) corresponding phase-angle bode plots of electrochemical impedance spectra of the pristine STON and CoPi/STON photoanodes deposited at 30 min and measured at 1.25 V vs. RHE under illumination, and (**c**) the corresponding Mott Schottky plots of STON and CoPi/STON photoanodes at 100 Hz.

**Figure 10 nanomaterials-13-00920-f010:**
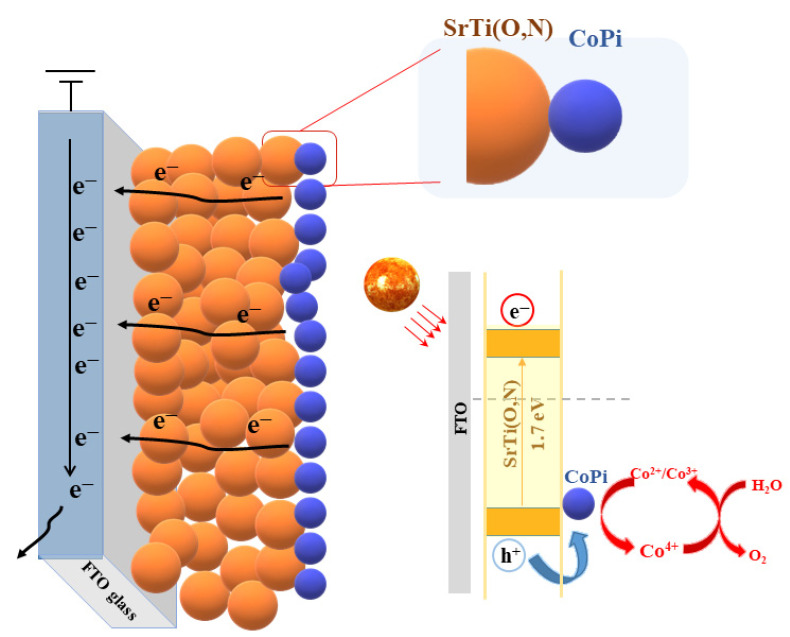
Schematic demonstration of charge transfer pathways in fabricated SrTi(O,N)/CoPi electrodes.

**Table 1 nanomaterials-13-00920-t001:** The EDX elemental analysis of the pristine STON, and CoPi modified STOM films before and after being used in the electrolysis process for 3 h.

Catalyst/Element	Sr Wt. %	Ti Wt. %	O Wt. %	N Wt. %	Co Wt. %	P Wt. %
STON	21.70	27.85	44.59	5.86	-	-
CoPi/STON (before)	19.01	27.06	48.81	3.86	0.22	1.04
CoPi/STON (after)	12.74	21.77	65.02	Not detected	0.12	0.35

**Table 2 nanomaterials-13-00920-t002:** The fitting impedance parameters of solution resistances (R_1_), constant phase element (Q_2_), charge transfer resistance (R_2_) and Warburg element obtained from the Nyquist plots under an equivalent circuit model are shown in the inset of Figure 9a.

Photoanode	R_1_/Ω	Q_2_ µF	R_2_ (kΩ)	W_2_ (KΩ·s^^−1/2^)
SrTi(O,N) dark	27.94	37.49	17.80	32.48
SrTi(O,N) light	28.99	71.63	14.90	8. 89
CoP/SrTi(O,N) dark	27.35	52.43	13.84	7.08
CoP/SrTi(O,N) light	28.1	119.9	6.99	5.93

## Data Availability

The data presented in this study are available on request from the corresponding author.

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
