# Peer review of "Photoelectrochemical Performance of Strontium Titanium Oxynitride Photo-Activated with Cobalt Phosphate Nanoparticles for Oxidation of Alkaline Water"

_nanomaterials, 2023, doi:10.3390/nano13050920_

Round 1

Reviewer 1 Report

In this submission (nanomaterials-2171938), Arunachalam* and coworkers synthesized a series of strontium titanium oxynitride (STON) via solid phase synthesis and assembled as a photoelectrode by electrophoretic deposition, and their morphological, optical properties and PEC performance for alkaline water oxidation are investigated. Further, cobalt-phosphate (CoPi) based co-catalyst was photo-deposited over the surface of the STON electrode to boost the PEC efficiency to give a photocurrent density of ~ 138 μA/cm at 1.25 V versus RHE. These studies confirm a viable approach to improve energy conversion efficiency by merging water oxidation catalysts with innovative perovskite-type oxynitride electrodes.

Overall, this work well demonstrates an example of CoPi/STON as an effective approach for boosting solar-assisted water oxidation reactions, which is of a certain novelty and might be suitable for Nanomaterials, if the author can address sufficiently the following considerations below:

1. "More importantly, it must be validated that the level of the CoPi loading depends mainly on the fabricated electrode's morphological features, time, and methodology of CoPi decoration." It is recommended to supplement the physicochemical characterization of the CoPi decorated STON/FTO photoanode with varied CoPi loading by varying the duration of photodeposition process in SI.

2. Ti 2p spectrum in Figure 4c is a mistake, please revise Figure 4.

3. "The maximum performances with a photocurrent density of 11.2 μA/cm2 at 1.25 V vs. RHE were recorded for the STON photoanodes with a thick layer of about 1.6 μm" and "In comparison, the PEC performance of CoPi/STON photoanode displayed more than 400 mV cathodic shift of the photocurrent onset potential and an enhancement in the photocurrent of nearly 26.0 μA/cm2 at 1.25V vs RHE which is more than four times higher compared to pristine STON electrode (6.0 μA/cm2 at 1.25 V vs RHE)" Please carefully check these data.

4. The saying of "It was found that both the STON and modified CoPi/STON photoanodes are quite stable with no signs of any decay during the electrolysis process." is inaccurate according to the data in Figure 8b.

5. For the References section, the format of references should be unified, such as Ref 11 and Ref. 12. Meanwhile, very few recent references on photocatalysts are cited, so it is suggested to update more recent literature, such as 10.1007/s10853-020-04971-2; 10.1039/D1CC02468J.

6. Please provide characterization data after long-term testing, including but not limited to SEM, XRD, etc.

7. Please carefully check and correct typos in MS, such as "Oxynitride-based photoanodes are regard as......"; "photo=deposition"; "Figure 3 shows the powder XRD pattern of the pristine SrTi(O,N)3 the powder......"; some SPACEs should be removed; etc.

8. The format of words in Figures should be consistent, including the font size, sample name, etc.

9. No Supplementary Information is provided in the first round of peer review.

Author Response

Reviewer #1

Comments and Suggestions for Authors

In this submission (nanomaterials-2171938), Arunachalam* and coworkers synthesized a series of strontium titanium oxynitride (STON) via solid phase synthesis and assembled as a photoelectrode by electrophoretic deposition, and their morphological, optical properties and PEC performance for alkaline water oxidation are investigated. Further, cobalt-phosphate (CoPi) based co-catalyst was photo-deposited over the surface of the STON electrode to boost the PEC efficiency to give a photocurrent density of ~ 138 μA/cm at 1.25 V versus RHE. These studies confirm a viable approach to improve energy conversion efficiency by merging water oxidation catalysts with innovative perovskite-type oxynitride electrodes.

 Overall, this work well demonstrates an example of CoPi/STON as an effective approach for boosting solar-assisted water oxidation reactions, which is of a certain novelty and might be suitable for Nanomaterials, if the author can address sufficiently the following considerations below:

Reply: many thanks to the reviewer for the valuable and constructive comments that improved the manuscript’s scientific quality.

  1. "More importantly, it must be validated that the level of the CoPi loading depends mainly on the fabricated electrode's morphological features, time, and methodology of CoPi decoration." It is recommended to supplement the physicochemical characterization of the CoPi decorated STON/FTO photoanode with varied CoPi loading by varying the duration of the photo deposition process in SI.

         Reply: many thanks for the reviewer’s suggestion. In this work, the PEC                performance of the photoanodes modified with CoPi deposited at different times was firstly screened by LSV under dark and light illumination, and then the physicochemical characterizations of SEM, EDX, XRD, and XPS were performed for the samples with the best performance. This is because of the analysis cost and the limited access to such characterization techniques.  However, in compliance with the reviewer’s suggestion, we used XRD to characterize the STON/FTO photoanode with CoPi nanoparticles that photo-deposited at different times and the result is provided in Figure S2 (SI). The XRD pattern in Figure S2 shows there are no discernible differences in the pattern of the CoPi/SrTi(O, N) electrode compared to the pristine SrTi(O, N) one after CoPi deposition, showing that the CoPi content is below the lower limit of XRD detection.

  1. Ti 2p spectrum in Figure 4c is a mistake, please revise Figure 4.

Reply: Sorry for truncating the curve labelling. Figure 4c is rectified in the revised manuscript.

  1. "The maximum performances with a photocurrent density of 11.2 μA/cm2 at 1.25 V vs. RHE were recorded for the STON photoanodes with a thick layer of about 1.6 μm" and "In comparison, the PEC performance of CoPi/STON photoanode displayed more than 400 mV cathodic shift of the photocurrent onset potential and an enhancement in the photocurrent of nearly 26.0 μA/cm2 at 1.25V vs RHE which is more than four times higher compared to pristine STON electrode (6.0 μA/cm2 at 1.25 V vs RHE)" Please carefully check these data.

Reply: many thank for the observation. The data are revised and the current value of SOTON is corrected and highlighted.

  1. The saying "It was found that both the STON and modified CoPi/STON photoanodes are quite stable with no signs of any decay during the electrolysis process." is inaccurate according to the data in Figure 8b.

Reply: We thank the reviewer for their care and precision. Yes, the current has been increasing by about 40% at the end of the electrolysis process, presumably due to the convection of evolved oxygen gas as well as the catalyst surface activation during the electrolysis process.  The text has been modified and highlighted on page 12.

  1. For the References section, the format of references should be unified, such as Ref 11 and Ref. 12. Meanwhile, very few recent references on photocatalysts are cited, so it is suggested to update more recent literature, such as 10.1007/s10853-020-04971-2; 10.1039/D1CC02468J.

Reply: Thank you for the comment. Ref. 11 and Ref 12 have been unified and one relevant suggested reference is cited in reference 13 in the revised manuscript. The second one is not relevant to current work as it is related to CO2 photoreduction.

  1. Please provide characterization data after long-term testing, including but not limited to SEM, XRD, etc.

Reply: We appreciate the reviewer’s suggestion to improve the manuscript.  In compliance with the reviewer’s request, we provide the crystal structure, elemental analysis and surface morphology using  XRD, EDX, and SEM analysis respectively for the  CoPi/STON photoanodes before and after the long-term stability test, and the results are presented in SI Figure S3 and Table 1 in the manuscript.

  1. Please carefully check and correct typos in MS, such as "Oxynitride-based photoanodes are regarded as......"; "photo=deposition"; "Figure 3 shows the powder XRD pattern of the pristine SrTi(O, N)3 the powder......"; some SPACEs should be removed; etc.

Reply: Thank you for your constructive comment. The typos have been corrected.

  1. The format of words in Figures should be consistent, including the font size, sample name, etc.

Reply: All the Figures formats have been revised and standardized to the best of our knowledge.

  1. No Supplementary Information is provided in the first round of peer review.

Reply: Sorry for not uploading the Supporting Information

Reviewer 2 Report

Nanomaterials (ISSN 2079-4991)

Manuscript

Title: Photoelectrochemical performance of strontium-titanium oxynitride photo-activated with cobalt phosphate nanoparticles for oxidation of alkaline water.

The authors discuss the photoelectrochemical activity of strontium titanium oxynitride (STON) and CoPi/STON. The article is interesting. However, I would have insisted more emphasis on the characterization of CoPi and CoPi/STON clusters in general.

Here are my remarks:

You wrote a particle size of 15 nm. A histogram of the CoPi size distribution is not shown. Also, the scale of the SEM images in Figure 1 is very small. It would be better to get better images like TEM which could confirm better verification of CoPi nanoparticles synthesized on STON.

EDS analysis doesn’t show many information will be better to add the quantifications of each element.

Why is it interesting to obtain a Perovskite-like structure? What was your objective?  Perovskites are not normally stable in water.

I wonder about the XPS characterization when it was done after Chronoamperometry? After 30 min of deposition? This point is not clear.

What happened to CoPi/STON after the Chronoamperometry, is it stable?

The current density at CA increases, do you still have CoPi deposition. Why this increase?

XPS analysis, N1s nitrogen shows an obvious change after CoPi. Can you explain why?

Harmonization of units on the graph "()", "/", ", "Figure 3, Figure 8, etc.

At the XRD Figure, the unlabeled peaks correspond to CoPi?

For ECSA, how did you get the specific capacitance? the ECSA  set-up should be added to the methodology.

To get the current density, which area do you use for ECSA or geometric area?

In Figure 6c, are the results from the front or back side?

In the schematic demonstration of charge transfer Figure 10. , the light shines from the front of the system, but you get better performance by shining from the back.

You wrote about charge carriers in CoPi cocatalysts, which are very efficient catalysts, improving hole trapping. And if you are using h+, why are you adding a hole trap (Na2SO3)?

Time-resolved spectroscopy, or transient absorption spectroscopy (TAS) that could confirm the charge carrier dynamics and the mechanism of water oxidation.

Check the LSV graphs in Figure 8. The current density before 0.4 RHEV started at high or negative current density.

What is the area of your Pt electrode?

For the EIS analysis, why did you set the AC voltage amplitude to 20nmV?

EIS measurements must be done under stable conditions, you apply 1.25V, but at chronoamperometry shows that you have a not stable system. Could you please explain?  

Author Response

Reviewer #2

Comments and Suggestions for Authors

Nanomaterials (ISSN 2079-4991)

Manuscript

Title: Photoelectrochemical performance of strontium-titanium oxynitride photo-activated with cobalt phosphate nanoparticles for oxidation of alkaline water.

The authors discuss the photoelectrochemical activity of strontium titanium oxynitride (STON) and CoPi/STON. The article is interesting. However, I would have insisted more emphasis on the characterization of CoPi and CoPi/STON clusters in general.

Here are my remarks:

1- You wrote a particle size of 15 nm. A histogram of the CoPi size distribution is not shown. Also, the scale of the SEM images in Figure 1 is very small. It would be better to get better images like TEM which could confirm better verification of CoPi nanoparticles synthesized on STON.

Reply: Thank you for your valuable comments. A histogram is added as an inset in Fig. 1d and the whole Figure has been modified in the revised manuscript to enhance the scale bar visibility. We are not able to provide TEM analysis this time because the accessibility for TEM machine is limited and it would take a long time to obtain the results.

2- EDS analysis doesn’t show much information will be better to add the quantifications of each element.

Reply: Thank you for your comments. We have included the EDX and elements quantification results for pristine STON and CoPi-modified STON before and after the stability test provided in Table 1 and Fig. S4 in SI in the revised manuscript.

3- Why is it interesting to obtain a Perovskite-like structure? What was your objective?  Perovskites are not normally stable in water.

Reply: Thank you for your valuable comments. Perovskite-based photoanodes offer an alternative to oxide-based anodes for visible-light absorption to produce H2 and O2 via water electro- and photo-splitting. The perovskite-relevant materials have the advantage of tuning the composition of ABO3 structure via complete or partial replacement of elements in A, B and O sites which enhance the conductivity, photon absorption, charge separation, oxygen vacancies, etc.….  The reviewer is right, some of the perovskites are not stable, but with such modification, the stability can be improved. For example, our research has shown that the activity and stability of OER have been improved when the O site is partially replaced by F ions (MA Ghanem, et al., Catalysts (2021) 11 1408). Moreover, this work and others show that the modification of  Perovskite-based materials with suitable co-catalysts demonstrated a new pathway to develop highly efficient and stable PEC water-splitting catalysts.

4- I wonder about the XPS characterization when it was done after Chronoamperometry? After 30 min of deposition? This point is not clear.

Reply: XRD characterization was performed for the pristine STON and after the 30 min deposition of CoPi. The text on pages 7 & 8 is revised to clarify this point.

5- What happened to CoPi/STON after the Chronoamperometry, is it stable? The current density at CA increases, do you still have CoPi deposition? Why this increase?

Reply: this is an interesting point and has been raised by reviewer# 1 (comments 4 &6). Yes, the current increased by about 40% at the end of the stability test and in compliance with this comment, we executed the surface morphology, elemental analysis and XRD characterizations analysis of the CoPi/STON electrode after being used in a stability test for 3 hours of electrolysis and the characterization results are reported in Fig S4 and Table 1. It found that no obvious changes have been observed for the CoPi/STON electrode in surface morphology or crystal structure after being used for three hours in the stability test. However, the elemental EDX analysis in Table 1 has shown a significant increase in oxygen wt. % and a decrease of Sr, Ti, Co, and P wt. % which could be related to the oxidation and leaching/mechanical effect of oxygen bubbles evolution during the electrolysis process. the text in the manuscript is revised and highlighted on pages 6, 12, and 13.    

6- XPS analysis, N1s nitrogen shows an obvious change after CoPi. Can you explain why?

Reply: This is a very good observation, many thanks to the reviewer. Yes, the ratio between the deconvoluted peaks of N 1s significantly changes after the deposition of CoPi nanoparticles. The peak area of the negatively charged N species is significantly decreased which could be related to the CoPi being preferentially deposited at the nitrogen negative site of the oxynitride substrate. The text on page 7 is modified to clarify this point.

7- Harmonization of units on the graph "()", "/", ", "Figure 3, Figure 8, etc.

Reply: both Figures are revised and the labels are corrected.

8- In the XRD Figure, the unlabeled peaks correspond to CoPi?

Reply: No diffraction fingerprint has been detected for CoPi by XRD, the unlabeled peaks correspond to the FTO substrate. Revised Figure 3 is included in the revised manuscript.

9-  For ECSA, how did you get the specific capacitance? the ECSA  setup should be added to the methodology.

Reply: sorry for missing the Supporting Information upload. More details for the estimation of the double-layer capacitance and the electroactive surface area are reported in the Supporting Information.

10- To get the current density, which area do you use for ECSA or geometric area?

Reply: We have used the geometric area to show current density.

11- In Figure 6c, are the results from the front or backside?

Reply: Back-side illumination

12- In the schematic demonstration of charge transfer Figure 10. , the light shines from the front of the system, but you get better performance by shining from the back.

Reply: sorry for the misrepresentation of the light illumination in the scheme. The photoanode was illuminated from the back side and the corresponding scheme is corrected.

13- You wrote about charge carriers in CoPi cocatalysts, which are very efficient catalysts, improving hole trapping. And if you are using h+, why are you adding a hole trap (Na2SO3)?

Reply: many thanks for this comment. The CoPi/STON films displayed an obvious enrichment of the photocurrent Reply and a reduction of onset potential in the presence of Na2SO3. The role of CoPi cocatalyst not only provides the active sites for the hydroxide oxidation but also acts as a hole sink and suppresses the charge recombination by timely consuming the photogenerated holes. While the Na2SO3 (electron donator) act as a hole scavenger and further suppress the recombination of charge carriers and improves the injection of photoinduced holes into the CoPi/STON/electrolyte. The text on pages 11 and 14 is modified and highlighted.  

14- Time-resolved spectroscopy, or transient absorption spectroscopy (TAS) that could confirm the charge carrier dynamics and the mechanism of water oxidation.

Reply: yes, it would be interesting to investigate the charge carrier dynamics and the mechanism of water oxidation at the CoPi/STON electrode using the above-mentioned techniques. However, these facilities are not currently available at our lab.  

15- Check the LSV graphs in Figure 8. The current density before 0.4 RHEV started at high or negative current density.

Reply: yes, we agree with the reviewer. The electrodes’ behaviour below 0.4 V RHE have not been investigated, but this can be due to the light photo effect where the electrodes showed a negative current value under dark condition while under light illumination the current increased to a +ve value due to the photoactivation of the added Na2SO3 hole scavenger.

16- What is the area of your Pt electrode?

Reply: Sorry for missing it, the Pt sheet counter electrode area equals 1.0 cm2. The text is modified in the experimental section on page 3.  

17- For the EIS analysis, why did you set the AC voltage amplitude to 20nmV?

Reply: In EIS analysis, the applied signal amplitude commonly can be in the range of 5 – 20 mV to ensure linearity. 

18- EIS measurements must be done under stable conditions, you apply 1.25V, but chronoamperometry shows that you have a not stable system. Could you please explain?  

Reply: yes, we agree with the reviewer. The current in chronoamperometry is gradually increased by about 40% over 3 hours of electrolysis, but impedance measurement takes only a few minutes during which the current does not significantly change and should be reasonably stable.    

Round 2

Reviewer 1 Report

My questions are well addressed, so it can be accepted now.

Author Response

Comment: My questions are well-addressed, so it can be accepted now.

Reply: Many thanks to the reviewer for the time being and for the valuable comments. 

Reviewer 2 Report

After my main remarks, I suggest the authors to check the electrochemical characterization again.   Figure 6, Figure 7a and Figure 8a.  Before 0.4V RHEV start at positive or negative current density.

In general, before publication the figures need to be improved (very small font, scale not visible, etc.).

Author Response

Comment 1: After my main remarks, I suggest the authors to check the electrochemical characterization again.   Figure 6, Figure 7a and Figure 8a.  Before 0.4 V RHEV start at positive or negative current density.

Reply: many thanks for the reviewer's care and precision. The voltammograms in the Figures mentioned above have been revised and the current measurements have been extended to a potential below 0.4 V vs. RHE and the zero current line has been added with all voltammograms now starting at a negative current density value. 

Comment 2: In general, before publication, the figures need to be improved (very small font, scale not visible, etc.).

Reply: The labels and fonts in the applicable Figures have been increased for more clarity.